# Protective Effects of a Strawberry Ellagitannin-Rich Extract against Pro-Oxidative and Pro-Inflammatory Dysfunctions Induced by a High-Fat Diet in a Rat Model

**DOI:** 10.3390/molecules25245874

**Published:** 2020-12-11

**Authors:** Ewa Żary-Sikorska, Bartosz Fotschki, Adam Jurgoński, Monika Kosmala, Joanna Milala, Krzysztof Kołodziejczyk, Michał Majewski, Katarzyna Ognik, Jerzy Juśkiewicz

**Affiliations:** 1Department of Microbiology and Food Technology, Faculty of Agriculture and Biotechnology, University of Science and Technology, Al. Prof. S. Kaliskiego 7, 85-796 Bydgoszcz, Poland; 2Division of Food Science, Institute of Animal Reproduction and Food Research of the Polish Academy of Sciences, Tuwima 10, 10-748 Olsztyn, Poland; b.fotschki@pan.olsztyn.pl (B.F.); a.jurgonski@pan.olsztyn.pl (A.J.); j.juskiewicz@pan.olsztyn.pl (J.J.); 3Institute of Food Technology and Analysis, Lodz University of Technology, Stefanowskiego 4/10, 90-924 Łódź, Poland; monika.kosmala@p.lodz.pl (M.K.); joanna.milala@p.lodz.pl (J.M.); krzysztof.kolodziejczyk@p.lodz.pl (K.K.); 4Department of Pharmacology and Toxicology, Faculty of Medicine, University of Warmia and Mazury, Warszawska 30, 10-082 Olsztyn, Poland; michal.majewski@uwm.edu.pl; 5Department of Biochemistry and Toxicology, Faculty of Biology, Animal Sciences and Bioeconomy, University of Life Sciences, Akademicka 13, 20-950 Lublin, Poland; katarzyna.ognik@up.lublin.pl

**Keywords:** rat model, gastrointestinal tract, high-fat-diet, strawberry extract, ellagitannins, nasutin A, urolithin A

## Abstract

Due to the demonstrated intestinal microbial transformation of strawberry ellagitannins (ET) into bioactive metabolites, in the current study on rats, we hypothesised that the dietary addition of a strawberry ET-rich extract (S-ET) to a high-fat diet (HFD) would attenuate disturbances in the redox and lipid status as well as in the inflammatory response. We randomly distributed 48 Wistar rats into six groups and used two-way analysis of variance (ANOVA) to assess the effects of two main factors—diet type (standard and high-fat) and ET dosage (without, low, and 3× higher)—applied to rats for 4 weeks. In relation to the hypothesis, irrespective of the dosage, the dietary application of ET resulted in the desired attenuating effects in rats fed a HFD as manifested by decreased body weight gain, relative mass of the epididymal pad, hepatic fat, oxidized glutathione (GSSG), triglycerides (TG), total cholesterol (TC), and thiobarbituric acid-reactive substances (TBARS) concentrations as well as desired modifications in the blood plasma parameters. These beneficial changes were enhanced by the high dietary addition of ET, which was associated with considerably higher concentrations of ET metabolites in the urine and plasma of rats. The results indicated that S-ET could be effectively used for the prevention and treatment of metabolic disturbances associated with obesity, dyslipidaemia, redox status imbalance, and inflammation.

## 1. Introduction

High-fat diets (HFD), including high-palm oil diets, applied to laboratory rodents are useful research models as the animals under such feeding regimens are attributed to experiencing physiological and metabolic disturbances that correspond to pathologically obese humans [1]. Undesired changes in lifestyle, including physical inactivity and the overconsumption and consumption of foods rich in fat, are factors leading to an overwhelming rise in the prevalence of overweight and obesity [2]. High dosages of dietary palm oil, which contains 50% saturated fatty acids, was reported to induce impaired glucose tolerance via diminished insulin sensitivity. Research demonstrated that even the treatment of mice with a short-term HFD, rich in saturated fatty acids, resulted in significant increase in visceral fat accumulation and in plasma leptin and pro-inflammatory cytokine levels as well as dyslipidaemic changes in the blood and liver [3]. Fruit and, in particular, various berries have been widely studied for their physiological action using in vitro and in vivo models; however, there is a need to determine which compounds are the most bioactive and which provide healing effects in a disease state [4].

The hypothesized health benefits connected to strawberry ingestion include the prevention and mitigation of inflammation, free-radical detrimental actions, liver steatosis, fat tissue development, blood dyslipidaemia, oxidative changes in low-density lipoproteins, and other disorders [5,6,7]. Recent data demonstrated that strawberry consumption led to the diminished progression of precancerous lesions and exerted chemoprevention in rodent tissues [8]. Many observations suggested that the potential chemoprevention of berries (including strawberries) may be exerted via a variety of combinations of type and amount of bioactive compounds; their quality and synergism; and the level of fruit processing, type of diet [8], and intestinal microbiota [9]. The latter appears to be of paramount importance as almost all berry compounds must undergo microbial transformation before entering the body’s circulatory system [9].

A study reported that strawberries are abundant in bioactive compounds with antioxidant properties, like ellagic acid (EA), ellagitannins (ET), anthocyanins, and vitamin C [10]. In strawberry fruit, the typical ET is agrimoniin, commonly found in other plants belonging to the *Rosaceae* family [11]. Research reported that ET, including agrimoniin, possesses antibacterial (e.g., against *H. Pylori*) [12], antihistaminic (e.g., potassium superoxide inhibition) [13], anti-inflammatory (e.g., neutrophil elastase inhibition) [14], antidiabetic (α-glucosidases activity inhibition) [15], and antioxidant (potent scavenging activity in H_2_O_2_, O_2_^−^, DPPH, and HClO assays) properties [16].

According to the current knowledge, the metabolism and absorption of EA and ET begin in the upper gastrointestinal tract, especially in the jejunum. The second phase of transformation takes place in the liver tissue, resulting in a metabolite range of conjugated forms that enter systemic circulation [17]. However, the majority of EA and ET consumed in the diet pass through the small intestine and enter the large gut, where they are the subject of microbial transformation leading to the formation of derivatives, including different types of urolithins and nasutins depending on the species [18].

These microbial metabolites are extensively absorbed in the large intestine and further transformed in the colonocytes and hepatocytes by phase II enzymes, yielding glucuronide, sulphate, and methyl derivatives, which appear in circulation at high nM to low µM concentrations [17,19]. Although ETs are not absorbed themselves, potent bioactivity was also attributed to urolithins as they were found in micromolar concentrations in body fluids and tissues after the ingestion of ellagitannin-containing products.

Giorgio et al. [20] reported that urolithin D selectively inhibited EphA2 phosphorylation, while Sanchez-Gonzales et al. [21] observed that urolithin A inhibited cell proliferation through the upregulation of p21 and induced apoptosis coupled to caspases 3 and 7 activation in prostate cancer cells. A strong attenuating in vitro effect of urolithin A on ox-LDL endothelial dysfunction was noted by Han et al. [22], which was mediated through miR-27 expression and the ERK/PPARγ pathway (extracellular signal regulated kinase/peroxisome proliferator–activated receptor gamma). Researchers reported that, in comparison to urolithins B and C, urolithin A exhibited stronger anti-inflammatory, e.g., the strongest inhibition of TNFα (tumor necrosis factor alpha) release and antioxidant activities [23,24,25].

Fotschki et al. [26,27] recently showed that raspberry ET metabolites were responsible for hypoglycaemic effects and improved the lipid profile in the plasma of rats fed a diet enriched with lard, and those beneficial effects were partly connected with the upregulation of fibroblast growth factor 19 (FGF19) and the decreased formation of secondary bile acids. In other work from that research team, a significant decrease in lipid peroxidation in several tissues, improvement in the serum and hepatic lipid profile, and enhanced anti-inflammatory effects followed greater amounts of ET metabolites appearing in the caecal digesta and body fluids of rats fed a high-fructose diet supplemented with strawberry extract rich in monomeric ellagitannins [28].

Jurgoński et al. [29] reported that the high-fat feeding regimen, besides changes in lipid metabolism and redox status, led to some unfavourable disturbances in large intestinal fermentative processes and microbial activity. Therefore, it could be expected that ET metabolite formation would be considerably affected by dietary high-fat treatment.

In the reported experiment on laboratory rats, the hypothesis was drawn that the health-promoting effects of the short-term dietary addition of strawberry extract rich in polyphenols were noticed with the HFD ingestion. These beneficial effects are also expected to be pronounced along with the intensified metabolism of ET-rich strawberry extract when added at a higher dosage.

## 2. Results

The E × D interaction showed that the highest BW gain was attributed to the F rats followed by the LEF and the HEF animals (in all cases, *p* < 0.05 vs. all other groups), and the lowest gain was noted in the control C rats (*p* < 0.05 vs. all groups except LE; Table 1). Irrespective of the extract dosage (without, low, or high), the high-fat treatment caused a significant decrease in the daily feed intake and the body lean tissue percentage (nuclear magnetic resonance analysis) as well as an increase in the body fat tissue percentage and the relative mass of the epididymal pad and the liver (*p* < 0.05 vs. standard diet). The dietary application of the strawberry extract at both dosages significantly decreased the relative mass of the epididymal pad, regardless of diet type (*p* < 0.05 vs. N).

The nature of the E × D interaction regarding hepatic fat percentage was that, among high-fat groups, it was the highest and the lowest in rats F and HEF, respectively. The lowest fat content in the liver was noted in the three groups fed standard diets (*p* < 0.05 vs. all groups fed HFD diets). The two-way ANOVA revealed that the high-fat treatment significantly decreased the GSH concentration, GSH/GSSG ratio, and PPARα expression in the liver, irrespective of the extract treatment (*p* < 0.05 vs. E; Table 2; Figure 1). The high extract dosage significantly increased the hepatic GSH, and both extracts increased the hepatic GSH/GSSG ratio and PPARα expression compared to the treatment N without extract addition.

In the PPARα case, the highest expression was noted upon the high extract dosage (*p* < 0.05 vs. N and low). The highest hepatic GSSG concentration was noted in the F group followed by the LEF and HEF groups, and the lowest values were noted in the LE and HE rats (*p* < 0.05 vs. all groups fed HFD diets). The hepatic GSSG concentration was comparable in the C and HEF rats (see significant interaction E × D, *p* < 0.001). The E × D was also significant in the case of the hepatic TBARS, TG, and TC concentrations (Table 2 and Figure 2). The F group excelled compared to all others with respect to those parameters (*p* < 0.05), while the lowest TBARS, TG, and TC concentrations were noted in the livers of rats fed the three normal-fat diets.

Two-way ANOVA showed a significant decrease in the plasma SOD and CAT activities as well as the plasma ACW capacity in the rats fed an HFD, regardless of the extract addition (Table 2). Irrespective of diet type, the highest SOD, CAT, and ACW values were noted in the extract high dosage treatment. The low treatment was higher than the N treatment with respect to plasma SOD and CAT activities (*p* < 0.05). In regard to the plasma FRAP, the E × D interaction showed the highest and the lowest values in the HE and F groups, respectively (*p* < 0.05 vs. all remaining groups). Additionally, among rats fed an HFD, the highest and lowest FRAPs were attributed to the HEF and F groups, respectively.

The highest plasma LOOH concentration was noted in the F rats (*p* < 0.05 vs. all other groups), while the lowest LOOH was in the LE group (*p* < 0.05 vs. F, LEF, and HEF); the plasma LOOH values were statistically comparable in the HE and HEF groups, while the difference between the LE and LEF groups was statistically significant (see E × D interaction, *p* = 0.003). With respect to the plasma ACL, the nature of E × D was that the highest values were noted in the LE, HE, and HEF rats (*p* < 0.05 vs. remaining groups) while the lowest ACL was in the plasma of F animals (*p* < 0.05 vs. all other groups).

The two-way ANOVA showed that the dietary high-fat treatment, regardless of extract addition, decreased the plasma HDL concentration and profile value as well as increased the plasma non-HDL, TG concentrations, and the value of the atherogenic coefficient (*p* < 0.05 vs. standard diet treatment; Table 3). Irrespective of diet type, the applied low and high dosages significantly decreased the plasma non-HDL concentrations, while the high dosage treatment significantly increased the HDL concentration and profile and decreased the value of the atherogenic coefficient (*p* < 0.05 vs. N). As shown by the E × D interaction, the value of the atherogenic index of plasma lg(TG/HDL) was the lowest in the three standard groups and the highest in the F rats.

The E × D interaction revealed that the highest plasma AST activity followed feeding with the F diet, while the lowest was the high dosage extract low-fat diet (*p* < 0.05 vs. all remaining groups and *p* < 0.05 vs. F and LEF groups, respectively; Table 4). The F rats excelled compared with all others for the plasma ALT (interaction *p* = 0.027). Significant interactions were also noted for the plasma TNF-α, IL-1β, and IL-6 concentrations. In all cases, the lowest values were observed in all rats fed standard diets, while the highest values were in the rats fed a HFD diet without extract (*p* < 0.05 vs. all groups).

Regardless of diet type, both extract dosages increased the plasma adiponectin level (*p* < 0.05 vs. treatment without extract). The high-fat treatment decreased the plasma adiponectin concentration, irrespective of extract addition (*p* < 0.05 vs. standard diet treatment). The highest daily excretion with the urine of urolithin A glucuronide was noted in HE animals (*p* < 0.05 vs. HEF, LE, and LEF), while the lowest was in the LE and LEF groups (*p* < 0.05 vs. HE, HEF; Table 5). With respect to urinal nasutin A glucuronide excretion, the highest value was in the HE group (*p* < 0.05 vs. all others), and the lowest was in the LEF one (*p* < 0.05 vs. HE and HEF rats).

The highest and the lowest daily nasutin A urinal excretions followed feeding with the HE and LEF diets, respectively (*p* < 0.05 vs. LE and LEF and *p* < 0.05 vs. HE and HEF). When urolithin A and total ET metabolites were considered, their daily excretion in urine was significantly higher in both groups fed a high E dosage compared with rats fed a low dosage (*p* < 0.05). In the plasma of LE and LEH rats, no nasutin A glucuronide was detected, and the latter group as well as the HEF group had no nasutin A in their plasma. The highest plasma concentrations of both mentioned ET metabolites were detected in HE animals. Both groups with high extract dosage had significantly higher values than the low dosage groups with respect to the plasma concentrations of DMEAG and the total ET metabolites (*p* < 0.05).

## 3. Discussion

The ET-rich strawberry extract used in the present study contained 82.3% phenolic compounds with a predominant content of ET (57.3%) and proanthocyanidins (23.9%) and a little content of EA and flavonols (0.2% and 0.9%, respectively). Nowicka et al. [10] reported that fresh strawberries may contain quite high total content of phenolic compounds, i.e., up to 2250 mg per kg. These discrepancies in the phenolic substance concentrations in the strawberry fruit are ascribed to factors including the plant type, climate, season, temperature, and the degree of ripeness when harvested [30].

The lines of in vitro and in vivo models were developed as valuable tools for estimating human absorption and metabolism. In vivo analyses of how food components are metabolized and enter the circulatory system to exert their biological effects provide more knowledge than the former options; however, they can be poor predictors of human reactions. Well-designed pharmacological and nutritional studies in rodents are often a fundamental part of metabolic and mechanistic assessments in the current biological science [31]. In the present experiment, with the aid of the body surface area normalization method [32] and the data reported by Nowicka et al. [10] for the high-phenolics strawberry fruit, the lower dietary treatment was assessed to correlate to a daily consumption of 0.40 kg fresh strawberries by an adult weighing 70 kg.

We assumed that dietary treatments with low extract dosage would represent a feasible amount of fresh high-phenolics strawberries consumed by humans while the high extract dosage could be achieved by dried fruit or could supplement preparations for consumers. The observed beneficial results upon the applied lower extract dosage argue in favour of whole fruit consumption as they are an easy, affordable, and excellent source of not only antioxidants but also minerals, vitamins, and soluble and insoluble fibres. The present study’s dietary treatment with the “faulty” high-fat palm oil diet in rats caused a sequence of metabolic disturbances, namely increased body weight gains (mainly due to the elevated fat tissue content), then hepatic steatosis, undesired changes in the redox balance, and lipid status in the liver and blood.

The relatively low lipotoxicity and atherogenic power of dietary palm oil has been reported but only in a well-balanced dietary environment, and the observed negative consequences of palm oil consumption are due to a dose–response relationship [1], as in the present study. In the present study, the additional palm oil in the diet resulted in increased release of adipose tissue-connected cytokines, i.e., TNF-α, IL-1β, and IL-6 along with the decreased concentration of adiponectin in the blood plasma. The detrimental consequences of dietary palm oil excess and its main component palmitic acid are partly ascribed to a lipotoxic effect, i.e., mitochondrial malfunction mediated by oxidative stress [1].

Various cytokines, such as those mentioned above, were recognized as potent activators of signal transducer and activator of transcription 3 (STAT3) signalling in multiple target organs [33]. For instance, IL-6-STAT3 signalling was clearly shown to link obesity, inflammation, and hepatic neoplastic changes [34]. The cytokines penetrating the liver are involved in the recruitment and activation of Kupffer cells, which are resident hepatic macrophages and cause the transformation of stellate cells to the myofibroblastic phenotype. TNF-α promotes the activation of I кB kinase, which activates NF-кB, a pro-inflammatory trigger that regulates inflammatory mediators. TNF-α antagonises adiponectin, an anti-inflammatory adipocytokine [35].

Considering that the ETs are rather poorly available as themselves when compared to other polyphenol classes, the obtained results were of promise as both applied dosages (low and three times higher) effectively reduced undesired changes in the body fat distribution and the hepatic and blood parameters in rats fed an HFD. According to the accepted hypothesis, some desired hepatic changes were intensified following the high vs. low dosages of the extract in a diet. Those enhanced beneficial effects were observed in certain parameters of the liver, including a lowered fat content, TBARS, GSSG concentrations, and plasma AST activity and increased GSH concentration and PPARα expression.

Research reported that dietary polyphenols may activate AMPK (AMP-activated protein kinase) via SIRT1 and LKB1, thus inhibiting ACC activity and decreasing the production of malonyl-CoA. As a result, FA β-oxidation is upregulated along with downregulated FA synthesis, thereby leading to hepatocyte lipid reduction [36]. In the present experiment, the dietary strawberry extract, irrespective of the dosage, exerted similar antioxidative and anti-inflammatory effects in the liver and plasma of rats fed an HFD.

With regard to the accepted hypothesis, the high extract dosage strengthened the beneficial changes in the line of plasma indicators: SOD, CAT, FRAP, ACW, ACL, and HDL (observed increase) and LOOH, AST, atherogenic coefficient, TNF-α, IL-1β, and IL-6 (observed decrease). The polyphenols are sometimes called a “double-edged sword” with both anti-ROS and pro-oxidative properties [37]. There were not any observations made suggesting that high ET-rich extract application may result in undesired pro-oxidative or pro-inflammatory changes.

Our own reports and those of other authors have shown that the main ET metabolites in rats are urolithin A, nasutin A, and their glucuronide conjugates, which may be found in the intestinal digesta, internal tissues, and body fluids [28,32,38,39]. The present study showed the highest urinal and serum ET metabolite levels in rats fed diets with the higher extract dosage. The nature of the ET metabolite action in the body of healthy and unhealthy states is under constant investigation, and the line of in vitro and in vivo studies showed the possible mechanisms leading to health-supporting effects in humans and animals.

For instance, [40] revealed in vitro that the observed strong anti-inflammatory potential of urolithin A in macrophages was associated with an increased autophagic flux and an inhibition of M1 (LPS) macrophage polarization, the latter channelled into the elevated production of NO, ROS, and pro-inflammatory cytokines. Other authors have shown attenuating in vitro effects of urolithin A on endothelial dysfunctions induced by oxidized LDL through the beneficial modulation of TNF-α, IL-6, endothelin 1, PPAR-γ, ICAM-1 (intercellular adhesion molecule 1), and MCP-1 (monocyte chemotactic protein 1) expression [17].

Considering the accepted hypothesis, the health-promoting changes were observed in all rats fed an HFD with the extract; however, for certain parameters, the beneficial changes were strengthened in the group fed with the higher extract dosage (e.g., a reduction in BW gain and the atherogenic coefficient). The subtle measurement of fat and lean tissues performed by the nuclear magnetic resonance (NMR) analysis revealed a statistical tendency toward an elevated lean tissue percentage and diminished fat tissue percentage (*p* = 0.063 and *p* = 0.094, respectively) in the rats fed diets with a high extract dosage.

Those observations corroborated the higher ET metabolite concentrations in the rat plasma and urine. The recent work of Xia et al. [41] reported that urolithin A, a metabolite of ET, prevented diet-induced obesity and metabolic dysfunctions in mice without any adverse effects. Those authors showed that urolithin A treatment increased the energy expenditure by enhancing thermogenesis in brown adipose tissue and by inducing the browning of white adipose tissue. The molecular explanation revealed that urolithin A acted as a mediator of an elevated level of triiodothyronine in brown adipose tissue.

## 4. Materials and Methods

### 4.1. Preparation of the Strawberry Extract

Strawberry press cakes were collected from a strawberry juice production line of the Alpex Company (Łęczeszyce, Poland) and dried at 70 ± 2 °C. After drying, the press cakes were separated via the use of appropriate screens into a seed fraction (diameter 0.5–1 mm) and a seedless fraction (diameter 1–3 mm). The raw polyphenol extracts were obtained from the seedless fraction via alcohol and acetone extraction. We placed 6 kg of the seedless fraction and 20 L of 65% ethanol in water in a stainless steel 30 L volume extractor. The mixture was left for 48 h at 20–25 °C and, then, was separated on a laboratory press resulting in 14.7 L of ethanol extract and 10.2 kg of wet pomace. The solvent was recovered by distillation, which provided 6 L of residue.

The residue was rich in polyphenols and contained 15% ethanol. We mixed 10.2 kg of wet pomace after the first extraction with 15 L of 65% acetone in water and placed the mixture in the extractor at 20 °C for 24 h. After the second extraction, 15 L of the acetone–ethanol extract was separated from 10 kg of pomace (wet weight) on a laboratory press. The resulting 10 kg of pomace was mixed with 10 L of water and pressed after 1 h to result in 11 kg of wet pomace and 8 L of acetone–ethanol–water extract. Both acetone extracts were joined (15 L and 8 L) and rectified with acetone and ethanol recovery, and this resulted in 6 L of residue containing c.a. 15% of ethanol.

The residue was joined with the residue from the first extraction. We filtered 12 L of extract containing 600 g of dry matter on a cellulose filter subjected to purification. The joined extracts were then purified on an adsorbent Amberlite XAD in a 20-L column with 15 L of the adsorption bed. The process consisted of sorbent conditioning as recommended by the bed manufacturer: adsorption of the polyphenols in the column bed with a flow speed 1 bed volume (BV)/h; washing the low molecular weight saccharides and ions off of the bed with the same flow rate and the use of 2 BV of 8% of ethanol in water; and successive desorption of the polyphenol fractions with the opposite flow direction, a flow rate of 0.2 BV/h, and increasing ethanol concentration, i.e., 30%—1 BV and 55%—until the desorption was completed.

During the desorption, 0.1 BV fractions were collected and analysed for polyphenols. The fractions of similar compositions were joined, concentrated, and freeze-dried. The following fractions were distinguished by the proportions of primary polyphenol groups: the ETs, the proanthocyanidins (PACs), the ACs, and the flavan-3-ols. The resulting lyophilized products were placed in polyethylene terephthalate containers and stored at −4 °C in the absence of light. The nutritional components and polyphenols of the individual products were determined.

### 4.2. Basic Chemical Composition

The dry matter, ash, crude protein, and crude fat were determined according to the official Association of Official Analytical Chemists (AOAC) methods 920.151, 940.26, 920.152, and 930.09, respectively [42]. The carbohydrate contents were determined using the following formula: carbohydrate = total solids − (protein + fat + ash).

### 4.3. The Polyphenol Contents in the Extract

The concentrations of ETs, EA, and flavanols in the extracts were determined following their dilution in methanol (1 mg/mL) using a HPLC (Knauer Smartline system with a photodiode array detector, Berlin, Germany) coupled with a Gemini C18 column (110 Å, 250 × 4.60 mm; 5 μm, Phenomenex, Torrance, CA, USA). Phase A was 0.05% phosphoric acid in water, phase B was 0.05% phosphoric acid in 80% acetonitrile, the flow rate was 1.25 mL/min, the sample volume was 20 μL, and the temperature was 35 °C.

The following gradient was applied: stabilization for 5 min with 4% phase B, 4–15% B for 5–12.5 min, 15–40% B for 12.5–42.5 min, 40–50% B for 42.5–51.8 min, 50–55% B for 51.8–53.4 min, and 4% B for 53.4–55 min. The following standards were used for the identification of the polyphenols: flavanols (quercetin-3-*O*-glucoside, kaempferol-3-*O*-glucoside, quercetin, kaempferol, and tiliroside), ellagic acid (all from Extrasynthese, Genay, France), *p*-coumaric acid (Sigma-Aldrich, St. Louis, MO, USA), and samples of the ETs, specifically bishydroxydiphenoyl-D-glucose and agrimoniin, which were obtained by semi-preparative HPLC as described by Sójka et al. [43]. The absorbances were measured at 280 nm (for *p*-coumaric acid, tiliroside, hexahydroxydiphenoyl-D-glucose, and agrimoniin) and 360 nm (for ellagic acid, quercetin, kaempferol, and kaempferol glycosides).

The concentration of proanthocyanidins (PACs) in the extracts was determined by the HPLC method following PAC breakdown in an acidic environment with an excess of phloroglucinol according to the methods of Kennedy and Jones [44]. The obtained breakdown products were separated using a Knauer Smartline chromatograph (Berlin, Germany) equipped with an UV–Vis detector (PDA 280, Knauer, Berlin, Germany) and a fluorescence detector (Shimadzu RF-10Axl, Kyoto, Japan), coupled with a Gemini C18 column (110 Å, 250 × 4.60 mm; 5 μm, Phenomenex, Torrance, CA, USA).

The separation conditions were previously described by Kosmala et al. [45]. The identification was performed at 280 nm using a UV–Vis detector and the following standards: (−)-epicatechin, (+)-catechin, (−)-epigallocatechin, and their respective phloroglucinol adducts. Quantification was conducted based on the peak areas that were registered using a fluorescence detector (excitation wavelength: 278 nm; emission wavelength: 360 nm). The standard curves of (−)-epicatechin and (+)-catechin and the (−)-epicatechin-phloroglucinol adduct were used to quantify the breakdown products of the terminal units and extender units, respectively. The chemical and polyphenolic compositions of the strawberry extracts are presented in Table 6.

### 4.4. Experimental Intervention

The experiment was conducted on 48 male Wistar rats weighing 177 ± 1.351 g, randomly assigned to one of six groups of eight rats each. The animals were maintained individually in metabolic cages under a stable temperature (21–22 °C), a 12-h light:12-h dark cycle, and a ventilation rate of 15 air changes per hour. The rats were used in compliance with the European guidelines for the care and use of laboratory animals (EU Directive 2010/63/EU), and the animal protocol was approved by the Local Institutional Animal Care and Use Committee (approval No.10/2018). For 4 weeks, the rats had free access to tap water and semi-purified diets, which were prepared and then stored at 4 °C in hermetic containers until the end of the experiment (details in Table 7).

The diets were modifications of a casein diet for laboratory rodents recommended by the American Institute of Nutrition [46]. The corn starch (C), and corn starch and strawberry ET-rich extract applied at two doses, low and high (LE and HE, respectively), groups received diets based on corn starch, whereas the high-fat dietary treatments had an additional 14% palm oil content at the expense of corn starch in the high-fat control (F) and in the groups with low and high addition of the strawberry extract (LEF and HEF, respectively). The high-palm oil diet was selected as a useful research dietary model as laboratory rodents under this feeding regimen were found to experience physiological and metabolic disturbances that correspond to pathologically obese humans [1]. Additionally, the effective amount of palm oil to induce the aforementioned disturbances in rats was assessed in our previous experiments [47,48].

The diets were modified with the strawberry extract dosages (low or high) added at the expense of corn starch. All diets had equilibrated amounts of dietary protein. The diets with lower dietary extract dosage were prepared to represent a feasible amount of fresh strawberries consumed by humans. Taking into account the data reported by Nowicka et al. [10] for high-phenolics strawberry fruit and the body surface area normalization method [32], the lower dietary treatment correlated to a daily consumption of 0.40 kg fresh strawberries by an adult weighing 70 kg. In the case of higher extract dosage, that amount was 1.20 kg of fresh fruit.

### 4.5. Sample Collection

We determined the individual feed consumption and body weight (BW) gain of the rats. The rats were deprived of feed overnight (10–12 h) prior to anaesthesia with a ketamine/xylazine solution in 0.9% NaCl (100/10 mg/kg BW). At termination of the experiment, the rats were weighed and anesthetized as mentioned above. After a laparotomy, blood samples were collected to heparinized tubes from the caudal vein and plasma was prepared by low-speed centrifugation (350× *g*, 10 min, 4 °C). The plasma samples were kept frozen at −70 °C until assayed. Subsequent to blood collection and euthanasia, the liver and periepididymal adipose pad were dissected and immediately weighed.

### 4.6. Hepatic Basic Analyses

Thiobarbituric acid-reactive substances (TBARS), which create lipid peroxidation, were determined in the liver tissue after storage at −70 °C. A procedure developed by Botsoglou et al. [49] was used in the assay, and the TBARS contents were determined spectrophotometrically at 532 nm and expressed in µg malondialdehyde per g of tissue. After storage of the liver, the reduced glutathione (GSH) and oxidized glutathione (GSSG) concentrations were determined using an enzymatic recycling method described by Rahman et al. [50]. The liver lipids were extracted according to Folch et al. [51]. Following extraction, the total cholesterol (TC) and triglyceride (TG) concentrations were determined enzymatically using commercial kits (Cholesterol DST, Triglycerides DST, Alpha Diagnostics Ltd., Warsaw, Poland).

### 4.7. Hepatic RNA Isolation and Quantitative RT-PCR PPARα Analysis

The total RNA was extracted from the liver samples using the TRI Reagent (Sigma-Aldrich, St. Louis, MO, USA) according to the manufacturer’s instructions. The quantity and quality of RNA were measured spectrophotometrically using a NanoDrop1000 (Thermo Fisher Scientific, Waltham, MA, USA) and agarose gel electrophoresis, respectively. cDNA was synthesized from 500 ng of total RNA using a High-Capacity cDNA Reverse Transcription Kit with RNase Inhibitor (Applied Biosystem, Waltham, MA, USA). Glyceraldehyde 3-phosphate dehydrogenase (Gapdh) was selected as a reference gene.

The levels of peroxisome proliferator-activated receptor α (PPARα) and GAPDH mRNA expression were analysed using Single Tube TaqManVR Gene Expression Assays (Life Technologies, Carlsbad, CA, USA). The expression of the reference gene GAPDH in the liver was stable among all experimental groups (with 95% confidence intervals; equivalent to *p* > 0.05). Amplification was performed using a 7900 HT Fast Real-Time PCR System under the following conditions: initial denaturation for 10 min at 95 °C, 40 cycles of 15 s at 95 °C, and 1 min at 60 °C. Each run included a standard curve based on aliquots of pooled liver RNA. All samples were analysed in duplicate. The mRNA expression levels of PPARα were normalized to GAPDH and multiplied by 10.

### 4.8. Blood Plasma Antioxidant Status Analyses

In the blood plasma, the antioxidant capacities of water-soluble and lipid-soluble substances (ACW and ACL, respectively) were determined by a photochemiluminescence detection method using a Photochem and the respective kits (ACW-Kit and ACL-Kit, Analytik Jena AG, Jena, Germany). In the photochemiluminescence assay, the generation of free radicals was partially eliminated through reactions with antioxidants present in the plasma samples, and the remaining radicals were quantified by luminescence generation. Ascorbate and Trolox calibration curves were used to evaluate ACW and ACL, respectively.

The redox status was also characterized by superoxide dismutase (SOD), catalase (CAT), ferric reducing antioxidant power (FRAP), and lipid hydroperoxide (LOOH) measurements. The activity of SOD was determined using Ransod and Ransel diagnostic kits, and CAT was determined by the enzymatic decomposition of hydrogen peroxide (H_2_O_2_). The total antioxidant potency was measured with a FRAP (Ferric Reducing Antioxidant Power) assay kit. The FRAP assay measures the antioxidant potential in sample colorimetrically at 594 nm through the reduction of ferric iron (Fe(III)) to ferrous iron (Fe(II)) by antioxidants present in the samples. The lipid hydroperoxide (LOOH) assay kit measures the lipid hydroperoxide (LOOH) directly utilizing the redox reactions with Fe(II) to produce Fe(III). The resulting ferric ions were detected colorimetrically (500 nm) using thiocyanate ions as the chromogen.

### 4.9. Blood Plasma Biochemical Analyses

The triglycerides (TG), total cholesterol (TC), the fractions of HDL cholesterol (HDL), and glucose plasma concentrations as well as the activity of aspartate and alanine aminotransferases (AST and ALT, respectively) were estimated using a biochemical analyser (Pentra C200, Horiba, Tokyo, Japan). An enzyme immunoassay (Cusabio Biotech Co., Ltd., Wuhan, Hubei, China) was used to determine the rat plasma levels of adiponectin (Rat adiponectin ELISA Kit). To measure insulin concentration, a validated rat insulin ELISA kit was used (Demeditec Diagnostics GmbH, Kiel, Germany). The plasma concentrations of IL-1β, IL-6, and TNF-α were determined using Thermo Fisher Scientific assays (ELISA kits, Waltham, MA, USA). The indexes of plasma were calculated using the following formulas: the atherogenic index of plasma (AIP) = lg(TG/HDL) and atherogenic coefficient (AC) = (TC-HDL)/HDL), with values for TG, TC, and HDL in mmol/L.

### 4.10. Quantification of ET Metabolites

The concentration of ET metabolites was determined in the urine and blood plasma. The urine was collected during the 24 h one day before termination of the study. The ET metabolites were identified by comparison of UV spectra with the available literature data [18,52] and additionally confirmed by the MS method described below. A urine sample (0.5 mL) was mixed with acetone (1 mL), sonicated for 10 min, and centrifuged (5 min, 10,000× *g*), and then, the supernatant was collected in a test tube. The procedure was repeated, and both supernatants were collected in a test tube and concentrated using a vacuum concentrator (ScanSpeed 40, Labogene, Lillerod, Denmark).

Next, the concentrated sample was dissolved in methanol (200 µL) and analysed by HPLC-ESI-MS using a Dionex UltiMate 3000 UHPLC and a Thermo Scientific Q Exactive series quadrupole ion trap mass spectrometer. The ET metabolites were separated using a Kinetex C18 column (110 Å, 150 × 2.1 mm; 2.6 μm, Phenomenex, Torrance, CA, USA) and a binary gradient of 0.1% formic acid in water (phase A) and 0.1% formic acid in acetonitrile (phase B) at a flow rate of 0.5 mL/min, as follows: stabilization for 1.44 min with 5% B, 5–15% B for 1.44 to 2.98 min, 15–40% B for 2.98–10.1 min, 40–73% B for 10.1–11.5 min, 73% B for 11.55–12.7 min, 73–5% B for 12.7–13.28 min, and 5% B for 13.28–18 min. The MS analysis was performed in negative ion mode under the following conditions: capillary voltage at +4 kV, sheath gas pressure at 75 arbitrary units, auxiliary gas at 17 arbitrary units, and scan range 120–1200 *m*/*z*. Urolithin-A, isolated from human urine by semipreparative HPLC, was used as a standard for the quantification of ET metabolites. The detailed procedure of urolithin-A isolation is described elsewhere [30].

To 0.5 mL of serum, 1 mL 100% acetone was added; the mixture was mixed, sonicated for 10 min, and centrifuged (900 rad/s). The procedure was repeated with 1.0 mL of 100% acetone, and the extracts were combined and evaporated to dryness under vacuum in a ScanSpeed 40 low speed centrifuge (Labogene, Lillerod, Denmark), lyophilized in an Alpha 1-2 LD plus freeze−dryer (Martin Christ Gefriertrocknungsanlagen GmbH, Osterode am Harz, Germany), solubilized in 0.2 mL of 100% methanol, and then analysed using UHPLC-MS (parameters as for urine).

### 4.11. Statistical Analysis

The results are expressed as means and pooled SEM except for the chemical composition of the strawberry extracts, which is expressed as the means ± SDs. A two-factor analysis of variance (ANOVA) was used to determine the effect of the extract additions (E; none, low, and high dosages), the diet type (D; corn starch and high-fat treatments), and the interaction between these two factors (E × D). If the analysis revealed a significant interaction (*p* ≤ 0.05), the differences among the respective treatment groups were then determined using Duncan’s post hoc test at *p* ≤ 0.05. The data were checked for normality prior to the statistical analyses. If the ANOVA assumptions were violated, the Kruskal–Wallis one-way ANOVA by ranks was used followed by Dunn’s post hoc test (*p* ≤ 0.05). The statistical analysis was performed using STATISTICA software, version 12.0 (StatSoft Corp., Krakow, Poland).

## 5. Conclusions

In conclusion, the dietary application of both dosages of strawberry extract rich in bioactive ET resulted in considerable alleviating effects in metabolic processes that were disturbed by an HFD applied to rats for four weeks. The experiment was focused on the redox balance, lipid status, and inflammatory response, and the health-promoting actions of strawberry ET were observed in all those physiological areas in the liver and blood plasma. The paramount beneficial changes associated with extract ingestion included decreased fat content; decreased GSSG, TG, and TC concentrations in the liver tissue; increased SOD, CAT, FRAP, ACL, and adiponectin values; and decreased concentrations of TNF-α, IL-1β, and IL-6 in the blood plasma.

The high dosage of extract strengthened, without any adverse effects, the health-promoting actions of strawberry extract via enhanced levels of ET metabolites circulating in the rat body. Dietary strawberry ET-rich extract supplementation may, thus, be considered an effective strategy in supporting dietary tools for the prevention and treatment of chronic diseases associated with obesity.

## Figures and Tables

**Figure 1 molecules-25-05874-f001:**
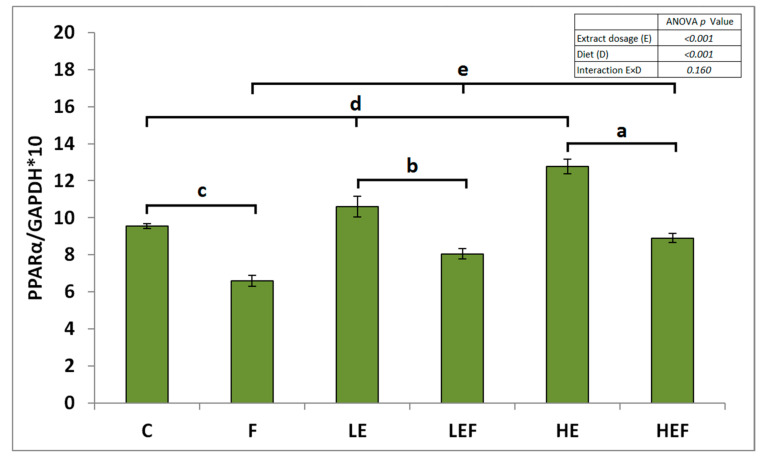
The mRNA expression of peroxisome proliferator-activated receptor α (PPARα) in the livers of rats fed experimental diets: the values are the means ± SEMs. To indicate a significant effect (at *p* < 0.05) of the extract dosage, bars were marked with the letters a, b, or c, whereas to present an effect of different types of diet, bars were marked with the letters d or e. C, control fed a diet with 53% corn starch; F, fed a diet with 14% pam oil added at the expense of corn starch; LE, fed a C-diet with a strawberry ET-rich extract at a low dosage of 0.24% of the diet; LEF, fed an F-diet with a strawberry ET-rich extract at a low dosage of 0.24% of the diet; HE, fed a C-diet with a strawberry ET-rich extract at a high dosage of 0.72% of the diet; and HEF, fed an F-diet with a strawberry ET-rich extract at a high dosage of 0.72% of the diet.

**Figure 2 molecules-25-05874-f002:**
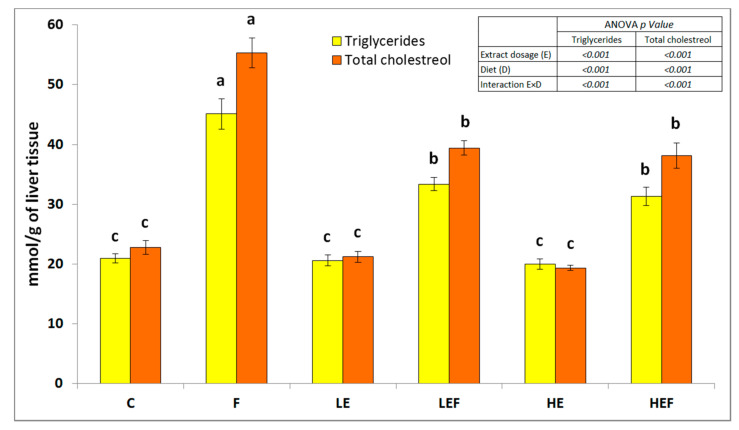
The hepatic level of cholesterol and triglycerides of rats fed the experimental diets: the values are the means ± SEMs. Mean values not sharing the same superscript letter within bars (a, b, or c) are different at *p* < 0.05. C, control fed a diet with 53% corn starch; F, fed a diet with 14% pam oil added at the expense of corn starch; LE, fed a C-diet with a strawberry ET-rich extract at a low dosage of 0.24% of the diet; LEF, fed an F-diet with a strawberry ET-rich extract at a low dosage of 0.24% of the diet; HE, fed a C-diet with a strawberry ET-rich extract at a high dosage of 0.72% of the diet; and HEF, fed an F-diet with a strawberry ET-rich extract at a high dosage of 0.72% of the diet.

**Table 1 molecules-25-05874-t001:** The growth parameters, body composition, and liver indices of rats fed the experimental diets *.

	Start BW	Gain	Intake	Body Composition			Liver	
	g	g	g/day	Fat ^1^, %	Lean ^1^, %	Epididymal Pad ^2^	Mass ^2^	Fat ^1^, %
Group (*n* = 8)								
C	277	41.3 ^e^	19.6	30.6	40.1	3.63	2.93	21.8 ^d^
F	276	68.0 ^a^	17.6	35.3	34.5	4.81	4.41	39.4 ^a^
LE	276	44.7 ^d,e^	19.5	30.7	39.3	2.62	2.87	21.3 ^d^
LEF	276	60.3 ^b^	17.8	34.9	36.5	4.41	4.18	36.1 ^b^
HE	276	47.2 ^d^	19.2	30.3	40.7	2.60	2.93	21.0 ^d^
HEF	277	53.0 ^c^	17.1	33.5	37.9	4.02	3.79	32.6 ^c^
SEM	0.513	1.539	0.205	0.366	0.453	0.145	0.110	1.185
Extract dosage (E)								
N (without)	276	54.7	18.6	32.9	37.3	4.22 ^a^	3.67	30.6
Low	276	52.5	18.6	32.8	37.9	3.52 ^b^	3.52	28.7
High	276	50.1	18.2	31.9	39.3	3.31 ^b^	3.36	26.8
*p* value	0.987	0.067	0.408	0.094	0.063	<0.001	0.134	0.005
Diet (D)								
Standard	276	44.4	19.4 ^a^	30.5 ^b^	40.0 ^a^	2.95 ^b^	2.91 ^b^	21.4
High-fat	276	60.4	17.5 ^b^	34.5 ^a^	36.3 ^b^	4.41 ^a^	4.13 ^a^	36.0
*p* value	0.966	<0.001	<0.001	<0.001	<0.001	<0.001	<0.001	<0.001
Interaction E × D								
*p* value	0.907	<0.001	0.838	0.324	0.163	0.318	0.130	0.036

* C, control fed a diet with 53% corn starch; F, fed a diet with 14% pam oil added at the expense of corn starch; LE, fed a C-diet with a strawberry ET-rich extract at a low dosage of 0.24% of the diet; LEF, fed an F-diet with a strawberry ET-rich extract at a low dosage of 0.24% of the diet; HE, fed a C-diet with a strawberry ET-rich extract at a high dosage of 0.72% of the diet; and HEF, fed an F-diet with a strawberry ET-rich extract at a high dosage of 0.72% of the diet. ^a,b,c,d,e^ Mean values within a column with unlike superscript letters were shown to be significantly different (*p* < 0.05); differences among the groups (C, F, LE, LEF, HE, and HEF) were indicated with superscripts only in the case of a statistically significant interaction E × D (*p* < 0.05). ^1^ Nuclear magnetic resonance (NMR) analysis; ^2^ g/100g BW; BW, body weight. SEM, pooled standard error of mean (standard deviation for all rats divided by the square root of rat number, *n* = 48).

**Table 2 molecules-25-05874-t002:** Redox status of the plasma and liver of rats fed the experimental diets *.

	Plasma						Liver			
	SOD	CAT	FRAP	LOOH	ACW	ACL	GSH	GSSG	GSH/GSSG	TBARS
	U/mL	U/mL	µmol/L	µmol/L	µg/mL	µg/mL	µmol/g	µmol/g		µg/g
Group (*n* = 8)										
C	40.8	17.7	183 ^b^	11.5 ^c,d^	4.06	17.5 ^b^	45.3	5.00 ^cd^	9.30	3.53 ^c,d^
F	35.2	11.6	134 ^d^	16.2 ^a^	3.26	14.2 ^c^	40.9	11.0 ^a^	3.79	5.59 ^a^
LE	44.6	22.3	189 ^b^	10.3 ^d^	4.11	21.5 ^a^	48.5	3.89 ^d^	12.8	3.46 ^c,d^
LEF	40.8	14.0	163 ^c^	13.2 ^b^	3.91	17.0 ^b^	44.0	8.00 ^b^	5.68	4.30 ^b^
HE	69.3	26.5	228 ^a^	11.1 ^c,d^	4.31	20.1 ^a^	52.8	4.43 ^d^	12.3	3.16 ^d^
HEF	58.3	16.1	193 ^b^	12.0 ^b,c^	4.16	19.8 ^a^	47.4	6.35 ^c^	7.82	3.97 ^b,c^
SEM	1.860	0.810	4.428	0.342	0.081	0.422	0.920	0.404	0.538	0.150
Extract dosage (E)										
N (without)	38.0 ^c^	14.6 ^c^	159	13.8	3.66 ^b^	15.9	43.1 ^b^	8.00	6.54 ^b^	4.56
Low	42.7 ^b^	18.1 ^b^	176	11.7	4.01 ^ab^	19.2	46.3 ^ab^	5.95	9.26 ^a^	3.88
High	63.8 ^a^	21.3 ^a^	211	11.6	4.24 ^a^	19.9	50.1 ^a^	5.39	10.1 ^a^	3.56
*p* value	<0.001	<0.001	<0.001	<0.001	0.004	<0.001	0.003	<0.001	<0.001	<0.001
Diet (D)										
Standard	51.6 ^a^	22.2 ^a^	200	11.0	4.16 ^a^	19.7	48.9 ^a^	4.44	11.5 ^a^	3.38
High-fat	44.8 ^b^	13.9 ^b^	164	13.8	3.78 ^b^	17.0	44.1 ^b^	8.45	5.76 ^b^	4.62
*p* value	<0.001	<0.001	<0.001	<0.001	0.007	<0.001	0.004	<0.001	<0.001	<0.001
Interaction E × D										
*p* value	0.121	0.068	0.019	0.003	0.103	0.004	0.959	<0.001	0.122	0.019

* C, control fed a diet with 53% corn starch; F, fed a diet with 14% pam oil added at the expense of corn starch; LE, fed a C-diet with a strawberry ET-rich extract at a low dosage of 0.24% of the diet; LEF, fed an F-diet with a strawberry ET-rich extract at a low dosage of 0.24% of the diet; HE, fed a C-diet with a strawberry ET-rich extract at a high dosage of 0.72% of the diet; and HEF, fed an F-diet with a strawberry ET-rich extract at a high dosage of 0.72% of the diet. ^a,b,c,d^ Mean values within a column with unlike superscript letters were shown to be significantly different (*p* < 0.05); differences among the groups (C, F, LE, LEF, HE, and HEF) were indicated with superscripts only in the case of a statistically significant interaction E × D (*p* < 0.05). SOD, superoxide dismutase; CAT, catalase; FRAP, ferric reducing antioxidant power; LOOH, lipid hydroperoxides; ACL, integral antioxidant capacity of lipophilic substances in plasma; ACW, integral antioxidant capacity of hydrophilic substances in plasma; GSH, reduced glutathione; GSSG, oxidized glutathione; TBARS, thiobarbituric acid reactive substances; SEM, pooled standard error of mean (standard deviation for all rats divided by the square root of rat number, *n* = 48).

**Table 3 molecules-25-05874-t003:** Biochemical indicators of the blood plasma lipid profile of rats fed the experimental diets *.

	TC	HDL	Non-HDL	HDL Profile	TG	Atherogenic Coefficient	Atherogenic Index of Plasma
	mmol/L	mmol/L	mmol/L	% of TC	mmol/L	(TC-HDL)/HDL	lg(TG/HDL)
Group (*n* = 8)							
C	1.88	0.933	0.946	49.6	1.32	1.02	0.146 ^c^
F	1.96	0.828	1.13	41.9	1.72	1.42	0.320 ^a^
LE	1.86	0.990	0.871	53.1	1.29	0.889	0.112 ^c^
LEF	1.92	0.864	1.06	45.1	1.62	1.23	0.269 ^b^
HE	1.88	1.06	0.823	56.3	1.41	0.780	0.114 ^c^
HEF	1.96	0.893	1.07	45.5	1.60	1.20	0.250 ^b^
SEM	0.021	0.019	0.021	0.879	0.043	0.041	0.017
Extract dosage (E)							
N (without)	1.92	0.880 ^b^	1.04 ^a^	45.7 ^b^	1.52	1.22 ^a^	0.233
Low	1.89	0.927 ^a,b^	0.963 ^b^	49.1 ^a,b^	1.45	1.06 ^a,b^	0.191
High	1.92	0.975 ^a^	0.947 ^b^	50.9 ^a^	1.51	0.992 ^b^	0.183
*p* value	0.805	0.049	0.026	0.001	0.763	0.003	0.211
Diet (D)							
Standard	1.87	0.993 ^a^	0.880 ^b^	53.0 ^a^	1.34 ^b^	0.898 ^b^	0.124
High-fat	1.95	0.862 ^b^	1.09 ^a^	44.2 ^b^	1.65 ^a^	1.29 ^a^	0.280
*p* value	0.102	<0.001	<0.001	<0.001	<0.001	<0.001	<0.001
Interaction E × D							
*p* value	0.969	0.719	0.557	0.428	0.550	0.821	0.016

* C, control fed a diet with 53% corn starch; F, fed a diet with 14% pam oil added at the expense of corn starch; LE, fed a C-diet with a strawberry ET-rich extract at a low dosage of 0.24% of the diet; LEF, fed an F-diet with a strawberry ET-rich extract at a low dosage of 0.24% of the diet; HE, fed a C-diet with a strawberry ET-rich extract at a high dosage of 0.72% of the diet; and HEF, fed an F-diet with a strawberry ET-rich extract at a high dosage of 0.72% of the diet. ^a,b,c^ Mean values within a column with unlike superscript letters were shown to be significantly different (*p* < 0.05); differences among the groups (C, F, LE, LEF, HE, and HEF) were indicated with superscripts only in the case of a statistically significant interaction E × D (*p* < 0.05). HDL, high-density lipoprotein; TC, total cholesterol; TG, triglycerides; SEM, pooled standard error of mean (standard deviation for all rats divided by the square root of rat number, *n* = 48).

**Table 4 molecules-25-05874-t004:** Biochemical indicators of the blood serum of rats fed the experimental diets *.

	AST	ALT	Adiponectin	TNF-α	IL-1β	IL-6	Insulin
	U/L	U/L	ng/mL	pg/mL	pg/mL	pg/mL	pmol/L
Group (*n* = 8)							
C	62.3 ^b,c^	23.1 ^b^	897	28.9 ^d^	88.6 ^d^	171 ^c,d^	27.6
F	81.7 ^a^	33.5 ^a^	708	82.8 ^a^	179 ^a^	421 ^a^	29.0
LE	62.6 ^b,c^	24.1 ^b^	931	28.5 ^d^	84.9 ^d^	149 ^c,d^	27.2
LEF	68.2 ^b^	26.7 ^b^	812	65.2 ^b^	155 ^b^	268 ^b^	28.8
HE	59.0 ^c^	22.5 ^b^	915	25.6 ^d^	88.3 ^d^	130 ^d^	26.7
HEF	64.7 ^b,c^	26.3 ^b^	841	48.2 ^c^	145 ^c^	219 ^b,c^	28.8
SEM	*1.478*	*0.790*	*16.65*	*3.536*	*5.756*	*17.98*	*0.545*
Extract dosage (E)							
N (without)	72.0	28.3	802 ^b^	55.8	134	296	28.3
Low	65.4	25.4	871 ^a^	46.9	120	208	28.0
High	61.8	24.4	878 ^a^	36.9	117	174	27.8
*p* value	0.001	0.037	0.043	<0.001	0.001	<0.001	0.921
Diet (D)							
Standard	61.3	23.3	914 ^a^	27.6	87.3	150	27.2
High-fat	71.5	28.9	787 ^b^	65.4	160	303	28.9
*p* value	<0.001	<0.001	<0.001	<0.001	<0.001	<0.001	0.136
Interaction E × D							
*p* value	0.016	0.027	0.211	0.003	0.004	0.015	0.962

* C, control fed a diet with 53% corn starch; F, fed a diet with 14% pam oil added at the expense of corn starch; LE, fed a C-diet with a strawberry ET-rich extract at a low dosage of 0.24% of the diet; LEF, fed an F-diet with a strawberry ET-rich extract at a low dosage of 0.24% of the diet; HE, fed a C-diet with a strawberry ET-rich extract at a high dosage of 0.72% of the diet; and HEF, fed an F-diet with a strawberry ET-rich extract at a high dosage of 0.72% of the diet. ^a,b,c,d^ Mean values within a column with unlike superscript letters were shown to be significantly different (*p* < 0.05); differences among the groups (C, F, LE, LEF, HE, and HEF) were indicated with superscripts only in the case of a statistically significant interaction E × D (*p* < 0.05). ALT, alanine transaminase; AST, aspartate transaminase; SEM, pooled standard error of mean (standard deviation for all rats divided by the square root of rat number, *n* = 48).

**Table 5 molecules-25-05874-t005:** Ellagitannin metabolite profile in the urine and plasma of rats fed the experimental diets *.

	Urinal Daily Excretion, µg/rat					Plasma, ng/mL			
	Urolithin A Glucuronide ^1^	Nasutin A Glucuronide ^2^	Urolithin A ^3^	Nasutin A ^4^	Total	Nasutin A Glucuronide ^2^	DMEAG ^5^	Nasutin A ^4^	Total
LE	5.40 ^c^	53.7 ^b,c^	17.0 ^b^	74.4 ^b,c^	150 ^b^	0.00	6.08 ^b^	1.18 ^b^	7.26 ^b^
LEF	2.60 ^c^	8.57 ^c^	1.90 ^b^	28.6 ^c^	41.7 ^b^	0.00	4.98 ^b^	0.00	4.98 ^b^
HE	629 ^a^	205 ^a^	1482 ^a^	321 ^a^	2637 ^a^	1.84 ^a^	30.0 ^a^	17.9 ^a^	49.8 ^a^
HEF	261 ^b^	105 ^b^	1401 ^a^	190 ^a,b^	1957 ^a^	0.964 ^b^	21.3 ^a^	0.00	22.3 ^a^
*SEM*	*83.63*	*18.42*	*159.6*	*35.25*	*246.6*	*0.157*	*1.991*	*1.444*	*3.293*
*p* value	0.017	<0.001	<0.001	0.009	<0.001	<0.001	<0.001	<0.001	<0.001

* LE, fed a C-diet with a strawberry ET-rich extract at a low dosage of 0.24% of the diet; LEF, fed an F-diet with a strawberry ET-rich extract at a low dosage of 0.24% of the diet; HE, fed a C-diet with a strawberry ET-rich extract at a high dosage of 0.72% of the diet; and HEF, fed an F-diet with a strawberry ET-rich extract at a high dosage of 0.72% of the diet. ^a,b,c^ Mean values within a column with unlike superscript letters were shown to be significantly different (*p* < 0.05 following a Kruskal–Wallis analysis of variance (ANOVA)). SEM, pooled standard error of mean (standard deviation for all rats divided by the square root of rat number, *n* = 32). ^1^ HPLC retention time (min) 6.74; MS [M − H]^−^ 403; MS/MS fragments 227.04, 113.02, 85.03; UV spectra (nm) 216, 279, 295, 304, 349. ^2^ HPLC retention time (min) 6.99; MS [M − H]^−^ 445; MS/MS fragments 445.19, 269.01, 113, 85.03; UV spectra (nm) 222, 279, 315, 367, 379. ^3^ HPLC retention time (min) 9.54; MS [M − H]^−^ 227; MS/MS fragments 113.02, 85.03; UV spectra (nm) 197, 218, 280, 307, 355. ^4^ HPLC retention time (min) 10.08; MS [M − H]^−^ 269; MS/MS fragments 269, 113, 85.03; UV spectra (nm) 227, 245, 284, 323, 389. ^5^ HPLC retention time (min) 7.50; MS [M − H]^−^ 505; MS/MS fragments 329.03, 113.02, 85.03; UV spectra (nm) 220, 249, 330, 351, 367.

**Table 6 molecules-25-05874-t006:** Chemical composition of the strawberry ellagitannins (ET) extract.

	Strawberry ET-Rich Extract (g/100 g)
Dry matter	91.3 ± 0.05
Ash	0.03 ± 0.04
Fat	-
Protein	1.83 ± 0.03
Other Components ^1^	7.17 ± 0.01
Total Polyphenols (HPLC-DAD)	82.3 ± 0.1
Ellagic Acid	0.2 ± 0.0
Ellagitannins (ET)	57.3 ± 0.1
Monomers	23.3 ± 0.1
Dimers	34.0 ± 0.1
Proanthocyanidins	23.9 ± 0.2
Flavonols	0.9 ± 0.0

The results are expressed as the mean ± SD, *n* = 2; ^1^ Low-molecular carbohydrates and structural components of plant cell walls, including dietary fibre.

**Table 7 molecules-25-05874-t007:** Composition of the group-specific diets *.

	Group (%)					
	C	F	LE	LEF	HE	HEF
Casein	20.0	20.0	20.0	20.0	20.0	20.0
Cellulose ^1^	5.0	2.0	5.0	2.0	5.0	2.0
Sucrose	10.0	10.0	10.0	10.0	10.0	10.0
Rapeseed Oil	7.0	7.0	7.0	7.0	7.0	7.0
Palm Oil	0	14.0	0	14.0	0	14.0
Mineral Mix ^2^	1.0	1.0	1.0	1.0	1.0	1.0
Vitamin Mix ^3^	3.5	3.5	3.5	3.5	3.5	3.5
Choline Chloride	0.2	0.2	0.2	0.2	0.2	0.2
DL-Methionine	0.3	0.3	0.3	0.3	0.3	0.3
Strawberry ET-Rich Extract	0	0	0.24	0.24	0.72	0.72
Corn Starch	53.0	42.0	52.76	41.76	52.28	41.28
Calculated Dietary Contents:						
Total Polyphenols	0	0	0.197	0.197	0.591	0.591
Ellagitannins (monomer–dimer ratio)	0	0	0.138 (40:60)	0.138 (40:60)	0.414 (40:60)	0.414 (40:60)
Proanthocyanidins	0	0	0.057	0.057	0.171	0.171

* C, control fed a diet with 53% corn starch; F, fed a diet with 14% pam oil added at the expense of corn starch; LE, fed a C-diet with a strawberry ET-rich extract at a low dosage of 0.24% of the diet; LEF, fed an F-diet with a strawberry ET-rich extract at a low dosage of 0.24% of the diet; HE, fed a C-diet with a strawberry ET-rich extract at a high dosage of 0.72% of the diet; and HEF, fed an F-diet with a strawberry ET-rich extract at a high dosage of 0.72% of the diet. ET—ellagitannins. ^1^ α-cellulose preparation was obtained from Sigma-Aldrich (No. C8002). ^2^ AIN-93G (Reeves 1997), g per kg mix: 357 g anhydrous calcium carbonate (40.04% Ca), 196 g monobasic potassium phosphate (22.76% P, 28.73% K), 70.78 g potassium citrate and tripotassium monohydrate (36.16% K), 74 g sodium chloride (39.34% Na, 60.66% Cl), 46.6 g potassium sulphate (44.87% K, 18.39% S), 24 g magnesium oxide (60.32% Mg), 6.06 g ferric citrate (16.5% Fe), 1.65 g zinc carbonate (52.14% Zn), 1.45 g sodium meta-silicate 9 9H_2_O (9.88% Si), 0.63 g manganous carbonate (47.79% Mn), 0.3 g cupric carbonate (57.47% Cu), 221.026 g powdered sucrose, and 0.275 g chromium potassium sulphate × 12H_2_O (10.42% Cr). The following components were added in mg per kg mix quantities: 81.5 mg boric acid (17.5% B), 63.5 mg sodium fluoride (45.24% F), 31.8 mg nickel carbonate (45% Ni), 17.4 mg lithium chloride (16.38% Li), 10.25 mg anhydrous sodium selenate (41.79% Se), 10 mg potassium iodate (59.3% I), 7.95 mg ammonium paramolybdate × 4H_2_O (54.34% Mo), and 6.6 mg ammonium vanadate (43.55% V). ^3^ AIN-93G (Reeves 1997), g per kg mix: 3.0 g nicotinic acid, 1.6 g Ca pantothenate, 0.7 g pyridoxine–HCl, 0.6 g thiamin-HCl, 974.655 g powdered sucrose, 0.6 g riboflavin, 0.2 g folic acid, 0.02 g biotin, and 2.5 g vit. B_12_ (cyanocobalamin, 0.1% in mannitol). The following components were added in IU per g quantities: 15.0 IU vit. E (all-rac-α-tocopheryl acetate, 500), 0.8 IU vit. A (all-trans-retinyl palmitate, 500,000), 0.25 IU vit. D_3_ (chole-calciferol, 400,000), and 0.075 IU vit. K-1 (phylloquinone).

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
