# Peer review of "Protective Effects of a Strawberry Ellagitannin-Rich Extract against Pro-Oxidative and Pro-Inflammatory Dysfunctions Induced by a High-Fat Diet in a Rat Model"

_molecules, 2020, doi:10.3390/molecules25245874_

Round 1

Reviewer 1 Report

The manuscript submitted by Å»ary-Sikorska and colleagues titled: “Protective effects of strawberry ellagitannin-rich extract against pro-oxidative and pro-inflammatory dysfunctions induced by high-fat diet in rats model” is aiming to assess the protective effect of strawberry extract dietary supplementation (ellagitannin-rich) on oxidation and inflammation as assessed by relevant biomarkers in rats.

The research question is interesting, and the methods and experimental design are well described. Results are presented clearly.

Concerns

Title: Title needs to be corrected as per the genitive, ie: rat model (since they used Wistars) instead of rats model”.

Authors state that sedentary lifestyle as well as high-fat diets lead to obesity. Obesity is a multifactorial outcome. From a nutritional perspective the current problem seems to be more along the lines of simple/refined sugars rather than fat per se even though high fat diets still pose a significant problem. In fact, from a metabolic perspective High Fructose Corn Syrup tends to be more lipogenic and obesity contributing as well as inducing metabolic deregulation and pathway abrogation more so than mere fat. Besides, it is important for the authors to distinguish among different types of lipid. For instance, monounsaturated fatty acids such as oleic acid found in olive oil and avocado are established to in fact be beneficial from a health standpoint, while a Mediterranean style diet which is not shy on fat does seem to produce weight maintenance and even loss. Other lipids such as omega-3 and -6 also are established to be beneficial. These points need to be raised in the discussion for sure as well as briefly in the introduction to provide a well-informed and scientifically accurate and sound context. Fatty acids do not necessarily elicit the same magnitude of oxidative and inflammatory responses. Authors mention palm oil which seemingly as well as cocoanut oil are more potent in producing unfavorable metabolic responses.

A general consideration with all extract studies is the translatability of results and the potential sustained results and any long-term consequences in humans. This also argues in favor of the whole fruit or food in general as opposed to isolates or extracts. The synergies seen by a vastly diverse mixture of bioactive compounds is more likely to effectively provide the optimal fine-tunning of signaling as opposed to an extract. This is something that needs to be discussed in the discussion section approaching the issue from a nutritional angle.

A useful review towards this and in connection with bioactive compounds is the following for authors to consider:

  1. Kristo AS, Klimis-Zacas D, Sikalidis AK. (2016) Protective Role of Dietary Berries in Cancer. Antioxidants. 5(4)37. doi:10.3390/antiox5040037.

Furthermore, it is interesting to address the relationship between the produced metabolites ending up in the colon and the microbiota. There is significant evidence to suggest that there is a strong relationship between the microbiota demography and risk for chronic disease as in T2DM and CVD.  A useful review is the following for authors to consider:

  1. Sikalidis AK, Maykish A (2020) The Gut Microbiome and Type 2 Diabetes Mellitus; discussing a complex relationship. Biomedicines. 8(1):8. doi.10.3390/biomedicines8010008.

The language (syntax, grammar and vocabulary use) would benefit the manuscript if improved. The reviewer recommends language editing by preferably an English native speaker.

Proofreading for typos and errors needs to be conducted.

Author Response

Dear Editor of Molecules,

Thank you very much for the review. We are sending our responses to the comments made regarding the manuscript: the manuscript Molecules 1009236, entitled “Protective effects of strawberry ellagitannin-rich extract against pro-oxidative and pro-inflammatory dysfunctions induced by a high-fat diet in a rat model” (the title has been changes as suggested by the Reviewer)

We would like to thank the reviewers for their very helpful comments regarding our manuscript. The comments are all valuable and have helped us to revise and improve our paper. After considering them carefully, we have made corrections that we hope will meet with approval. The main corrections in the paper (indicated in the manuscript in the red font) and our responses to the reviewers’ comments are as follows:

Reviewer: 1

Title: Title needs to be corrected as per the genitive, ie: rat model (since they used Wistars) instead of rats model”.

Answer: It has been corrected as suggested.

Authors state that sedentary lifestyle as well as high-fat diets lead to obesity. Obesity is a multifactorial outcome. From a nutritional perspective the current problem seems to be more along the lines of simple/refined sugars rather than fat per se even though high fat diets still pose a significant problem. In fact, from a metabolic perspective High Fructose Corn Syrup tends to be more lipogenic and obesity contributing as well as inducing metabolic deregulation and pathway abrogation more so than mere fat. Besides, it is important for the authors to distinguish among different types of lipid. For instance, monounsaturated fatty acids such as oleic acid found in olive oil and avocado are established to in fact be beneficial from a health standpoint, while a Mediterranean style diet which is not shy on fat does seem to produce weight maintenance and even loss. Other lipids such as omega-3 and -6 also are established to be beneficial. These points need to be raised in the discussion for sure as well as briefly in the introduction to provide a well-informed and scientifically accurate and sound context. Fatty acids do not necessarily elicit the same magnitude of oxidative and inflammatory responses. Authors mention palm oil which seemingly as well as cocoanut oil are more potent in producing unfavorable metabolic responses.

Answer: The relevant information has been added to the Introduction and Discussion sections. We agree to the statement as for more potent lipogenic effects of high-fructose treatments vs high-fat one. In the series of own experiments the metabolic action of bioactive constituents of strawberries were scrutinized when added to the standard, high-fructose and finally to high-fat diets provided to Wistar rats. The present paper is one of the effects of these experiments.

A general consideration with all extract studies is the translatability of results and the potential sustained results and any long-term consequences in humans. This also argues in favor of the whole fruit or food in general as opposed to isolates or extracts. The synergies seen by a vastly diverse mixture of bioactive compounds is more likely to effectively provide the optimal fine-tunning of signaling as opposed to an extract. This is something that needs to be discussed in the discussion section approaching the issue from a nutritional angle.

Answer: Thank you for those suggestions. We have tried to add appropriate sentences into the manuscript.

A useful review towards this and in connection with bioactive compounds is the following for authors to consider:

  1. Kristo AS, Klimis-Zacas D, Sikalidis AK. (2016) Protective Role of Dietary Berries in Cancer. Antioxidants. 5(4)37. doi:10.3390/antiox5040037.

Furthermore, it is interesting to address the relationship between the produced metabolites ending up in the colon and the microbiota. There is significant evidence to suggest that there is a strong relationship between the microbiota demography and risk for chronic disease as in T2DM and CVD.  A useful review is the following for authors to consider:

  1. Sikalidis AK, Maykish A (2020) The Gut Microbiome and Type 2 Diabetes Mellitus; discussing a complex relationship. Biomedicines. 8(1):8. doi.10.3390/biomedicines8010008.

Answer: Thank you for those suggestions. We used the proposed scientific articles to emphasize the role of the intestinal microflora in the metabolism of food components and the impact on the health of the body.

The language (syntax, grammar and vocabulary use) would benefit the manuscript if improved. The reviewer recommends language editing by preferably an English native speaker.

Proofreading for typos and errors needs to be conducted.

Answer: the text has been checked by an English native speaker and proofreading has been conducted.

I hope you will find the revisions satisfactory and I and all co-authors look forward to hearing your response to the revised manuscript. I hope that our manuscript could be published in your respectable Journal.

Reviewer 2 Report

This manuscript is interesting and the results of interest. However, the manuscript is very dense and does not read well. The thoughts are not well connected and rather just smashed together. The paper needs extensive revision for English language. Please see few suggestions below. Introduction: Line 37: replace among others with as well as Line 39: High-fat diet (HFD) applied to… Use HFD thereafter Line 44: have been widely Line 46: Were all the positive effects cited in this sentence results of ne single study (ref. 5)? Line 49 & 57: replace like with such as Line 60-61: “These microbial metabolites – do you mean urolithins and nasutins? Line 66-69: Again, all these effects were reported in one study (ref. 10)? Line 89: delete notably and faulty Line 97: Never start a sentence with an Arabic number. Spell it out. Line 98: L not l Line 99: h not hours Line 126: use abbreviations of ellagitannins and ellagic acid. You already defined the abbreviation in the intro 2.4 Animal study should be called experimental design or intervention Were the animals acclimated? If so, for how long? Did the animals start their diets on day 1? Line 168: Provide rationale for palm oil and its dose Line 172-174: How does the dose in each diet translate to humans? Section 2.5 in Methods should be divided into other subheadings Line 211: GADPH Confidence intervals and effect sizes should be reported along with p-values Tables are not effective and hard to understand. You should use graphs instead Line 411: 3 kg of fresh strawberries per day is not feasible to be consumed by humans Paragraphs in the discussion section are extremely long and should be broken down

Author Response

Dear Editor of Molecules,

Thank you very much for the review. We are sending our responses to the comments made regarding the manuscript: the manuscript Molecules 1009236, entitled “Protective effects of strawberry ellagitannin-rich extract against pro-oxidative and pro-inflammatory dysfunctions induced by a high-fat diet in a rat model” (the title has been changes as suggested by the Reviewer)

We would like to thank the reviewers for their very helpful comments regarding our manuscript. The comments are all valuable and have helped us to revise and improve our paper. After considering them carefully, we have made corrections that we hope will meet with approval. The main corrections in the paper (indicated in the manuscript in the red font) and our responses to the reviewers’ comments are as follows:

Reviewer: 2

This manuscript is interesting and the results of interest. However, the manuscript is very dense and does not read well. The thoughts are not well connected and rather just smashed together. The paper needs extensive revision for English language. Please see few suggestions below. Introduction: Line 37: replace among others with as well as Line 39: High-fat diet (HFD) applied to… Use HFD thereafter Line 44: have been widely Line 46: Were all the positive effects cited in this sentence results of ne single study (ref. 5)? Line 49 & 57: replace like with such as Line 60-61: “These microbial metabolites – do you mean urolithins and nasutins? Line 66-69: Again, all these effects were reported in one study (ref. 10)? Line 89: delete notably and faulty Line 97: Never start a sentence with an Arabic number. Spell it out. Line 98: L not l Line 99: h not hours Line 126: use abbreviations of ellagitannins and ellagic acid. You already defined the abbreviation in the intro 2.4 Animal study should be called experimental design or intervention

Answer: the text has been checked by an English native speaker and proofreading has been conducted. Thank you for some suggestions for improvement.

Were the animals acclimated? If so, for how long? Did the animals start their diets on day 1?

Answer: In our Animal Laboratory we keep the mother herd of Wistar rats, so the young growing rats are raised, grow up, and then enter the experiment in one building, in similar environment. Yes, the animals started their diets on day 1.

Line 168: Provide rationale for palm oil and its dose

Answer: The appropriate information has been added. The high-palm oil diet was selected as an useful research dietary model as the laboratory rodents under such feeding regimen are attributed with most physiological and metabolic disturbances as pathologically obese human being. Additionally, the effective amount of palm oil to induce the aforementioned disturbances in rats was assessed in our previous experiments.

Line 172-174: How does the dose in each diet translate to humans?

Answer: Those information has been provided in the Discussion section.

Section 2.5 in Methods should be divided into other subheadings Line 211: GADPH Confidence intervals and effect sizes should be reported along with p-values Tables are not effective and hard to understand. You should use graphs instead Line 411: 3 kg of fresh strawberries per day is not feasible to be consumed by humans Paragraphs in the discussion section are extremely long and should be broken down

Answer: We agree that for some readers the graph are more “digestible” and “understandable”, therefore some results have been presented in that form; but not all – we think that the results of two-way ANOVA analyses are well informative when provided in Tables as in the present manuscript, so please accept both graphs and tables. The Discussion section has been shortened as suggested.

I hope you will find the revisions satisfactory and I and all co-authors look forward to hearing your response to the revised manuscript. I hope that our manuscript could be published in your respectable Journal.

Sincerely,

Reviewer 3 Report

Comments on Żary-Sikorska et al.

The manuscript by Żary-Sikorska et al. described the beneficial effects of the strawberry ellagitannin-rich extract against oxidative and inflammatory conditions induced by high-fat diet in rats. The hypothesis and clear execution are the main strengths of this study. However, there are some points that need to be clarified before publication.

The introduction section has been descriptive. It is difficult to point out the background of the study, and limitations, as well as research questions. Please rewrite it.

Line 83, please confirm “FGRF19 factor” correct? Or it is FGF19 growth factor?

Methods are exclusively described in detail.

Line 211, “Gapdh” should be GAPDH.

Line 227, please adjust subscript for “H2O2”.

Line 298, please mention which assay was used to measure plasma concentrations of IL-1β, IL-6 and TNF-α? Is this ELISA?

Although authors included some inflammatory molecules (such as IL-1β, IL-6 and TNF-α), their downstream signaling pathways (such as STAT3, NFκB, etc.) are not addressed. Antioxidant signaling pathway (such as AMPK) is also not addressed. Authors are encouraged to include these two mains signaling cascades in the revised version.

In the whole manuscript there too many short forms (particularly in the results section) that hinder the understanding of the context. Some of them are not elaborated such as PET.

Author Response

Dear Editor of Molecules,

Thank you very much for the review. We are sending our responses to the comments made regarding the manuscript: the manuscript Molecules 1009236, entitled “Protective effects of strawberry ellagitannin-rich extract against pro-oxidative and pro-inflammatory dysfunctions induced by a high-fat diet in a rat model” (the title has been changes as suggested by the Reviewer)

We would like to thank the reviewers for their very helpful comments regarding our manuscript. The comments are all valuable and have helped us to revise and improve our paper. After considering them carefully, we have made corrections that we hope will meet with approval. The main corrections in the paper (indicated in the manuscript in the red font) and our responses to the reviewers’ comments are as follows:

Reviewer 3

The introduction section has been descriptive. It is difficult to point out the background of the study, and limitations, as well as research questions. Please rewrite it.

Answer: The Introduction section has been rewritten as suggested.

Line 83, please confirm “FGRF19 factor” correct? Or it is FGF19 growth factor?

Answer: Thank you, it was our mistake, it should be FGF19.

Methods are exclusively described in detail.

Line 211, “Gapdh” should be GAPDH.

Answer: corrected as suggested.

Line 227, please adjust subscript for “H2O2”.

Answer: corrected as suggested.

Line 298, please mention which assay was used to measure plasma concentrations of IL-1β, IL-6 and TNF-α? Is this ELISA?

Answer: Yes, ELISA kits. Corrected as suggested.

Although authors included some inflammatory molecules (such as IL-1β, IL-6 and TNF-α), their downstream signaling pathways (such as STAT3, NFκB, etc.) are not addressed. Antioxidant signaling pathway (such as AMPK) is also not addressed. Authors are encouraged to include these two mains signaling cascades in the revised version.

Answer: Thank you for that suggestions; unfortunately we are not able to perform such analyses due to the lack of biological materials. In the next projects we will do our best to perform those downstream signalling pathways. In the text those mechanisms have been described: It has been reported that various cytokines, such as TNF-α, IL-1β, IL-6, have been recognized as potent activators of signal transducer and activator of transcription 3 (STAT3) signalling in multiple target organs. Then for instance, IL-6-STAT3 signalling has been clearly shown to link obesity, inflammation, and hepatic neoplastic changes. The cytokines penetrating the liver are involved in the recruitment and activation of Kupffer cells, which are resident hepatic macrophages, and cause the transformation of stellate cells to the myofibroblastic phenotype. TNF-α promotes activation of I кB kinase, which activates NF-кB, a pro-inflammatory trigger that regulates inflammatory mediators. Moreover, TNF-α antagonices adiponectin, an anti-inflammatory adipocytokine.

In the whole manuscript there too many short forms (particularly in the results section) that hinder the understanding of the context. Some of them are not elaborated such as PET.

Answer: It has been checked and improved as suggested.

I hope you will find the revisions satisfactory and I and all co-authors look forward to hearing your response to the revised manuscript. I hope that our manuscript could be published in your respectable Journal.

Sincerely,

Round 2

Reviewer 2 Report

The manuscript is much improved. Please see below for two minor and one major point.

Minor points:

The abbreviation of HFD is inconsistent throughout the manuscript. Please use HFD, not HfD.

"Research reported that ET, including agrimoniin, possesses antibacterial (e.g., against H. Pylori, E. coli), antihistaminic (e.g., potassium superoxide inhibition), anti-inflammatory (e.g., neutrophil elastase inhibition), antidiabetic (α-glucosidases activity inhibition), and antioxidant (potent scavenging activity in H2O2, O2, DPPH, and HClO assays) properties [data provided in review 11]." This not best practice. Authors should provide an original article citation for each property claimed.

Major point:

An explanation for the strawberry extract dose should be provided in the methods section. You can still discuss it in the discussion section.

Having said that, the authors should provide a single equivalent dose of fresh strawberry consumption for each dose. A range does not make sense. And again, 3 kg of strawberry fruit per day is not feasible!

Author Response

Dear Editor of Molecules,

Thank you very much for the review. We are sending our responses to the comments made regarding the manuscript: the manuscript Molecules 1009236, entitled “Protective effects of strawberry ellagitannin-rich extract against pro-oxidative and pro-inflammatory dysfunctions induced by a high-fat diet in a rat model”

We would like to thank the reviewers for their very helpful comments regarding our manuscript. The corrections in the paper (indicated in the manuscript in the red font) and our responses to the reviewers’ comments are as follows:

Reviewer: 2

The manuscript is much improved. Please see below for two minor and one major point.

Minor points:

The abbreviation of HFD is inconsistent throughout the manuscript. Please use HFD, not HfD.

Answer: It has been corrected.

"Research reported that ET, including agrimoniin, possesses antibacterial (e.g., against H. Pylori, E. coli), antihistaminic (e.g., potassium superoxide inhibition), anti-inflammatory (e.g., neutrophil elastase inhibition), antidiabetic (α-glucosidases activity inhibition), and antioxidant (potent scavenging activity in H2O2, O2, DPPH, and HClO assays) properties [data provided in review 11]." This not best practice. Authors should provide an original article citation for each property claimed.

Answer: It has been corrected. We provide an original article citation for each property claimed.

Major point:

An explanation for the strawberry extract dose should be provided in the methods section. You can still discuss it in the discussion section.

Having said that, the authors should provide a single equivalent dose of fresh strawberry consumption for each dose. A range does not make sense. And again, 3 kg of strawberry fruit per day is not feasible!

Answer: Thank you for those suggestions The relevant information regarding the strawberry extract dose has been provided in the Methods section and corrected in the Discussion section.

Taking into account the data from a scientific article in which 90 varieties of strawberries were analyzed with regard to the content of polyphenols, after conversion, the lower dietary dose of the extract correlated with the daily consumption of 0.40 kg of fresh strawberries by an adult weighing 70 kg. In the case of the higher extract dose, it was 1.20 kg of fresh fruit.

I hope you will find the revisions satisfactory.

Sincerely,

Reviewer 3 Report

The author response is satisfactory and can accepted for publication.

Author Response

Dear Reviewer of Molecules,

Thank you very much for the review of the manuscript: the manuscript Molecules 1009236, entitled “Protective effects of strawberry ellagitannin-rich extract against pro-oxidative and pro-inflammatory dysfunctions induced by a high-fat diet in a rat model”

Reviewer: 3

The author's response is satisfactory and can be accepted for publication.

We would like to thank the reviewers for their very helpful comments regarding our manuscript.

Sincerely,